# Recent Progress in Lanthanide-Doped Inorganic Perovskite Nanocrystals and Nanoheterostructures: A Future Vision of Bioimaging

**DOI:** 10.3390/nano12132130

**Published:** 2022-06-21

**Authors:** Gowri Manohari Arumugam, Santhosh Kumar Karunakaran, Raquel E. Galian, Julia Pérez-Prieto

**Affiliations:** 1Instituto de Ciencia Molecular (ICMol), University of Valencia, Catedrático José Beltrán, 2, Paterna, 46980 Valencia, Spain; gowrimanohari1987@gmail.com; 2State Key Laboratory of Optoelectronic Materials and Technologies, Nanotechnology Research Center, School of Materials Science & Engineering, Sun Yat-sen University, Guangzhou 510275, China; sanstechno@gmail.com

**Keywords:** inorganic perovskite, lanthanide-doped nanocrystals, upconversion photoluminescence, nanoheterostructure

## Abstract

All-inorganic lead halide perovskite nanocrystals have great potential in optoelectronics and photovoltaics. However, their biological applications have not been explored much owing to their poor stability and shallow penetration depth of ultraviolet (UV) excitation light into tissues. Interestingly, the combination of all-inorganic halide perovskite nanocrystals (IHP NCs) with nanoparticles consisting of lanthanide-doped matrix (Ln NPs, such as NaYF_4_:Yb,Er NPs) is stable, near-infrared (NIR) excitable and emission tuneable (up-shifting emission), all of them desirable properties for biological applications. In addition, luminescence in inorganic perovskite nanomaterials has recently been sensitized via lanthanide doping. In this review, we discuss the progress of various Ln-doped all-inorganic halide perovskites (LnIHP). The unique properties of nanoheterostructures based on the interaction between IHP NCs and Ln NPs as well as those of LnIHP NCs are also detailed. Moreover, a systematic discussion of basic principles and mechanisms as well as of the recent advancements in bio-imaging based on these materials are presented. Finally, the challenges and future perspectives of bio-imaging based on NIR-triggered sensitized luminescence of IHP NCs are discussed.

## 1. Introduction

Nowadays, IHP NCs are considered prospective materials not only for photovoltaic applications but also for nanophotonics and nonlinear optics. Moreover, these nanomaterials have steadily attracted the research communities because of their unique physical properties such as high photoluminescence quantum yield (PLQY), tuneable PL and defect tolerance [1,2]. In particular, lead halide perovskite NCs are especially attractive for high-performance solar cells, lasers, photodetectors and light-emitting diodes (LEDs) owing to their superior optical properties including high PLQY, large absorption cross-section and tuneable emission throughout the entire visible spectrum [3,4,5,6,7], though these perovskites have poor stability, which limit their general application. IHP NCs usually exhibit linear optical properties under UV and visible light. It is important to note that near-infrared (NIR) photon upconversion is a unique process that converts two or more NIR photons into one of a higher energy, thus leading to anti-Stokes emissions. It is very difficult to achieve nonlinear upconversion (UC) emissions in IHP materials because of their low multiphoton efficiency and lack of intermediate energy levels [8,9,10,11].

Nanoparticles comprising an inorganic matrix such as NaYF_4_ and NaYbF_4_ with two or more lanthanide ions as dopants such as Yb/Tm, Yb/Er, etc., present higher stability than that of IHP NCs as well as higher multiphoton absorption efficiency [12]. The NIR-excitation of Ln NPs using a continuous-wave laser as an excitation source yielded UC emissions in the NIR-to-UV range. Ln NPs have generally been used in biological applications such as bio-imaging and phototherapy owing to the considerable penetration depth of NIR light into biological systems [13,14,15,16]. However, the fluorescence emissions of Ln NPs have fixed wavelengths as well as non-tuneable emission colours due to limited energy levels of lanthanide ions, which restricted their usage in multiplexed bioimaging or bioassays [17,18]. The wavelength tuning of Ln NP emissions requires select Ln ions and suitable doping concentrations. 

Although the optoelectronic applications of IHP NCs have been well explored, their bio-applications are limited by the lead toxicity and their low stability under humidity, light and heat. Clearly, the major drawback of the perovskites is their poor stability, which accumulates decomposition. Perovskites can cause toxicity in biological applications since they degrade when exposed to water due to their ionic nature [19,20,21]. However, the superior optical properties of IHP NCs provide several advantages to be used in the field of bioimaging. Nevertheless, there are many issues, such as the stability of these materials under physiological conditions, their integration in biomolecules during circulation, their epigenetic interaction and their possible toxicity during degradation processes, to be clarified before real life implementation [22]. Interestingly, the combination of perovskites and Ln NPs can solve the abovementioned issues through the formation of nanoheterostructures with superior optical properties. Tuning the UC emission wavelengths under NIR light excitation is a promising strategy [23,24,25]. In addition, the combination of IHP NCs and Ln NPs achieved UC emissions, and it can also endow perovskite NCs with long-term stability [26]. 

The aim of this work is to review the unique properties of nanoheterostructures based on the interactions between IHP NCs and Ln NPs as well as of those of LnIHP NCs. Here, we present a systematic discussion on basic principles and mechanisms as well as on the recent advancements in bioimaging based on these materials and the challenges for facing future developments in bioimaging. 

## 2. Metal Halide Perovskite Nanocrystals

Perovskite materials generally have a chemical formula of ABC_3_, in which A and B represent the cations of dissimilar sizes and C denotes the anions such as oxygen and halogens. The ideal perovskite structure is cubic symmetrical, which consists of corner sharing BC_6_ octahedra as a backbone with cuboctahedral voids occupied by A-cations, as depicted in Figure 1 [27]. If the A-ion is small or the B-ion is large, the tolerance factor decreases to <1, which favours orthorhombic, rhombohedral or tetragonal structures rather than a cubic structure.

Hybrid perovskites, such as methylammonium/formamidium lead halides (MA/FAPbX_3_) have attracted the interest of researchers because of their excellent performance in optoelectronic devices. This is based on their superior properties: tuneable bandgap, high light absorption capability, long carrier diffusion length and inexpensive raw materials for their preparation. However, they have been affected by instability issues not only in solar cells but also in other applications, such as in LEDs and bioimaging. Inorganic perovskites have been introduced into optoelectronic devices to overcome these issues [28,29,30]. Inorganic perovskite oxide, lead-free inorganic perovskites and inorganic perovskites with mixed halides have also been developed in past decades [31,32,33,34,35,36,37,38]. All-inorganic halide perovskites have also received great attention from various fields such as solar cells [39,40], lasers [41,42], LEDs [43,44], water splitting [45,46], etc. because of their high performance and stability. 

### 2.1. All-Inorganic Halide Perovskites (IHP)

According to the literature, oxide-based perovskites are the most actively studied materials in the perovskite family due to their excellent magnetic, ferro-electric and superconductive properties [47]. Interestingly, the first halide-based perovskite structures were obtained by Moller in caesium lead halides (CsPbX_3_) [48]. The photoconductive properties of halide-based perovskites are tuneable by varying the halide components, which helps to achieve different spectral responses. They have interesting optical properties, including tuneable emission, defect tolerance, quantum confinement effect and high PL quantum efficiency. Remarkably, the emission of CsPbX_3_ (X = Cl, Br, I or their mixture) NCs can be tuned from 400 nm to 700 nm, as shown in Figure 2 [1,49].

### 2.2. Ln-Doped Inorganic Halide Perovskites (LnIHPs)

Lanthanide ions have unique properties that come from their 4f electrons and make them an attractive tool for optical applications owing to their large quantum numbers (*n* = 4, 1 = 3) with rich spectroscopic properties [51]. Interestingly, they have advantageous optical properties such as sharp band emissions, large Stokes/anti-Stokes shifts, high luminescent lifetimes and excellent photostability [52,53]. These properties are desirable for their applications in lighting and displays [54], bioimaging [55], therapy [56] and sensing [57]. Furthermore, Ln-doped NPs can serve as multifunctional platforms for bio-applications either by doping or surface functionalization. Ln doping in hybrid perovskites also plays a key role in their optoelectronic properties. For example, MAPbI_3_ with lanthanides has been considered a desirable candidate for improving the permanence of perovskite solar cells. The incorporation of various lanthanide ions (Ln^3+^ = Ce^3+^, Eu^3+^, Nd^3+^, Sm^3+^ or Yb^3+^) into perovskite films largely enhances their performance, such as the efficiency and stability of the device, which is attributed to an enlarged grain size and crystallinity [20]. 

Interestingly, the inorganic perovskite NCs of CsPbX_3_ (X = Cl, Br and I) exhibited efficient and narrow 4f–4f emissions through sensitization by lanthanide ions, which achieved tremendous success in optoelectronics. Apart from normal Stokes spectral shifting behaviour, the CsPbCl_3_ NCs with Yb^3+^ as a dopant exhibited excellent NIR PL with QY of 200% due to quantum cutting phenomena. Beyond that, the IHP also possessed UC photoluminescence with the presence of desirable Ln ions and this review article is mainly focused on UC phenomenon. The crystal lattice of inorganic lead halide perovskites is suitable for lanthanide ion doping due to their octahedral coordination (CN = 6). In particular, CsPbX_3_ NCs showed strong absorption of visible light and intense emission, which can be attributed to their excitonic transitions along with their desirable charge transportation, and hence, they are considered ideal hosts for lanthanide ion doping [58].

Reinhard et al. [59] synthesized Yb^3+^-doped Rb_2_MnCl_4_ perovskite crystals to yield up-conversion luminescence (UCL) under NIR excitation. The crystals exhibited very intense yellow-orange UCL when excited at 15 K. The UC process comprises a sequence of ground state and excited-state absorptions under an excitation with 10 ns pulses. The UC mechanism is cooperative, and both Yb^3+^ and Mn^2+^ ions in the crystals can participate in the process. The UC in Rb_2_MnCl_4_:Yb^3+^ with a linear arrangement of Yb^3+^-Cl^−^-Mn^2+^ proved to be three orders of magnitude, which is more efficient than that of the Yb^3+^-(Br^−^)_3_-Mn^2+^ arrangement of face-sharing octahedra in CsMnBr_3_:Yb^3+^. However, three loss processes were observed at temperatures above 50 K: one was intrinsic for Rb_2_MnCl_4_, whereas the others were related to the presence of Yb^3+^ (thermal quenching and non-radiative relaxation). All of these loss processes are governed by basic physical laws and prevent materials containing Yb^3+^ and Mn^2+^ from being used as an upconverter at room temperature. 

Later on, Beurer et al. [60] prepared Tm^2+^-doped CsCaI_3_ and RbCaI_3_ single crystals and compared their properties, such as absorption, light emission and UCL. Both compounds possessed multiple emissions under an excitation at 21,834 cm^−1^ in the 10 to 300 K temperature range but differed in the dominant UC processes. It can be demonstrated that a slight chemical variation between CsCaI_3_:Tm^2+^ and RbCaI_3_:Tm^2+^ caused drastic effects on their emissive properties. A relatively modest distortion of TmI_6_ octahedron in RbCaI_3_:Tm^2+^ suggests that a reduction in relevant energy gaps between excited states generally influences the competition between radiative and nonradiative relaxation processes. 

The combination of CsCaI_3_:Tm^2+^ with a low multi-phonon rate constant and RbCaI_3_:Tm^2+^ with an efficient UC process can possibly yield an efficient UC phosphor. On the one hand, it requires a smaller distortion of coordination octahedron than in RbCaI_3_:Tm^2+^ to prevent a reduction in energy gap. On the other hand, the barycentre of (4f)^12^(5d)^1^ states at low energies in RbCaI_3_: Tm^2+^ can enable efficient nonradiative relaxation at the ^2^F_5/2_ crystal-field levels. The mixed crystals of Tm^2+^-doped RbCaI_3_ and CsCaI_3_ can tune the energy gaps to optimal values.

Understanding the impact of dopant ions in host materials on energy transitions via the UC process is crucial for further extending the properties and applications of UC materials. In this regard, Gong et al. [61] reported the Er-doped PbTiO_3_ (PTO) perovskite nanofibers as model systems for exploring the effects of tetragonality and polarization on their UCL properties. A clear emission enhancement was observed in the UC green band at 523 nm and in the red band at 656 nm of Er-doped PTO perovskite nanofibers when compared with those in Er-doped BaTiO_3_ or PTO particles. This enhancement is mainly attributed to UC processes assisted by low-energy phonons. These results pave a way for further understanding the energy transition processes between lanthanide dopants and host perovskite oxide matrices in UC processes and for extending the applications of UC materials. 

A single-band pure UC emission is beneficial for enhancing colour purity and bioimaging. Interestingly, Wu et al. [62] reported a successful achievement of single-band red UC emission in 2016 from Yb^3+^/Er^3+^ co-doped KMgF_3_ perovskite NCs. These NCs were prepared by means of a non-equivalent substitution strategy, in which Ln^3+^ ions could aggregate as supported by density functional theory (DFT) calculations and UC dynamic processes. The single-band emission under a laser excitation at 976 nm proved to be independent of dopant concentration and pump power, as shown in Figure 3. The aggregation of Ln^3+^ ions and the strong cross-relaxation between them in the KMgF_3_ matrix played important roles for the occurrence of a single-band red emission.

Concomitantly, Ge et al. [63] synthesized the Er-doped perovskite single-crystals of NaNbO_3_ nanorods using a hydrothermal method with different doping concentrations. They demonstrated the successful incorporation of Er^3+^ ions into the B site of Nb^5+^ in the ABO_3_ perovskite structure and then into the A site of Na^+^ by increasing the Er doping concentration. High-resolution transmission electron microscopy (HRTEM) confirmed the single-crystal features of NaNbO_3_ nanorods. These nanorods doped with 0.5 wt% of Er^3+^ exhibited a strong green emission and weak red emission, and the studies on the dependence of emission with a laser power corroborated the contribution of two photons. Based on their luminescent properties, these nanorods can be applied in novel multifunctional devices as well as in bioimaging. 

Although UC nanostructures play a vital role in bioimaging applications, the enhancement of the red/green UC emission ratio is still challenging. In 2018, Arumugam et al. [64] prepared RbPbI_3_:Er^3+^,Yb^3+^ nanowires and combined them with surface plasmons to improve the red UC emission at 652 nm which was slightly higher than that of green emission at 548 nm (R/G ratio of 1.068). The UC emission and the mechanism shown in the partial energy level diagram of Er^3+^ and Yb^3+^ are illustrated in Figure 4. Excitation at 980 nm promotes Yb^3+^ ions to the ^2^F_5/2_ level which resonantly conveys its energy to nearby Er^3+^ ions through energy transfer up-conversion (ETU) process and fosters them from ground state to ^4^I_11/2_ state. Most of the populations relax down to the level of ^4^I^13/2^ via non-radiative relaxation process. The populations at ^4^I_11/2_ and ^4^I_13/2_ levels absorb a second incident photon to reach ^4^F_7/2_ and ^4^F_9/2_ levels via excited state absorption (ESA) process. Hence, the populations from ^2^H_11/2_/^4^S_3/2_ and ^4^F_9/2_ levels relax down to the ground state of ^4^I_15/2_ radiatively, resulting in bright red emission with weak-to-moderate green emissions. 

The red emission was about eight times greater when introducing an oleate complex (to become a surface ligand) in the formation of the RbPbI_3_:Er^3+^,Yb^3+^ nanowires (NWs). In addition, the decoration of a UC system with AuNPs as surface plasmons greatly improved the red emission, reaching a R/G ratio of 26:1. The reduced green emission lifetime of NWs was consistent with resonance energy transfer from NWs to surface plasmons of Au NPs. By contrast, the addition of surface plasmons accumulated a longer lifetime of the red band emission. The red UC emission enhancement was caused by the addition of surface plasmons, which was assisted by a surface plasmon resonance (SPR) band at 660 nm. Therefore, RbPbI_3_:Er^3+^,Yb^3+^/Au NPs are suitable candidates for bioimaging due to their desirable lifetime and enhanced red UC emission.

## 3. Lanthanide-Doped Matrices (NaLnF_4_ Matrices, Ln NPs)

UC is an interesting phenomenon of bioimaging as it is a more efficient process than that of two-photon absorption and high harmonic generation. In past decades, numerous bio-probes have been reported, such as fluorescent proteins, dyes and quantum dots, although they are not suitable for life science applications. 

UC nanostructures are considered promising materials for bioprobes [65], and in particular, Er^3+^/Yb^3+^ co-doped NaYF_4_ has been recognized as an efficient UC system although it presents some undesirable background radiation because of its prominent green emission with a lower signal–noise ratio. These drawbacks have encouraged investigations into other UC perovskite matrices. Compared with perovskite oxide materials, the inorganic halide perovskites are suitable for bioimaging because of their excellent photo-stability and higher chemical durability [59]. Host lattices with heavier halides are beneficial for the stabilization of dopant ions [60]. Inorganic halide perovskites are considered promising UC materials for bioimaging because of their adjustable crystal structure, optical stability, resistance to photo-bleaching and photo-blinking, spectral distinguishability and chemical durability [66]. 

Nanoparticles consisting of lanthanide-doped matrices such as NaYF_4_:Yb,Er NPs have attracted the interest of research communities due to their advantageous properties such as narrow band gap emission, reasonable optical stability and high chemical stability when compared with traditional luminescent materials, e.g., organic dyes and NCs. Moreover, Ln NPs have been widely used in the field of biology owing to their deep penetration into biological tissues without any damage and high signal-to-noise ratio under NIR excitation. Doping of matrices with lanthanides is the most attractive tool for their optical applications because of large quantum numbers (*n* = 4, l = 3) of lanthanide ions and rich spectroscopic properties [51]. Lanthanides are mostly stable in the +3 oxidation state except for Ce^4+^, Tb^4+^ and Yb^2+^ ions. In addition, the size and morphology of Ln NPs play key roles in biomedical applications [67,68]. 

In early NIR-to-visible UCL bioimaging investigations, it was difficult to achieve tissue penetration depths in the scale of millimetres. However, Yin et al., reported UCL imaging for the first time with considerable tissue depth (a penetration depth of 1 cm) using a luminescent probe of NaYF_4_:Yb,Er NPs in nude mice [69]. Concomitantly, Jing et al., compared the UCL imaging of pork muscle tissues at different depths (0–1 cm) through injections of polymer-modified NaYF_4_:Yb,Er and KMnF_3_:Yb,Er. For the former, the image was detected at a depth of about 0.5 cm, whereas KMnF_3_:Yb,Er exhibited a very strong red emission, which was detected at a tissue depth of 1 cm [70]. Xiang et al., have reported the importance of antigen-loaded Ln NPs in labelling and stimulating dendritic cells (DCs), and the Ln NP-labelled DCs achieved high-sensitivity in vivo UCL imaging [71].

Subsequently, Hesse et al. [72] reported the rapid preparation of sub-10 nm level pure hexagonal (β-phase) NaYF_4_-based Ln NPs using a simple one-pot method, in which therminol 66 was used as a co-solvent and monodispersed Ln NPs were obtained in very short reaction times. The UCL properties of these NPs were tuned by varying the dopant concentrations (Nd^3+^ and Yb^3+^ as sensitizers, and Er^3+^ as an activator). The enhancement in UCL intensity was observed in Ln NPs with optimized concentrations of sensitizer and activator ions as well as coating with inert/active shell. The UCL spectrum of core β-NaYF_4_:Yb^3+^/Er^3+^ 20/2 % Ln NPs in cyclohexane exhibited three intense bands centred at *λ* = 525 (^2^H_11/2_→ ^4^I_15/2_ transition, G1), 545 (^4^S_3/2_→^4^I_15/2_ transition, G2) and 660 nm (^4^F_9/2_→^4^I_15/2_ transition, R) under an excitation of 976 nm.

The excitation of conventional Ln NPs such as NaYF_4_:Yb^3+^/Er^3+^(Tm^3+^) at 980 nm caused overheating and damage of living tissues with a reduction in luminescence due to water absorption at 980 nm. Interestingly, the incorporation of Nd^3+^ ions into Ln NPs shifted the excitation wavelength to 808 nm, thus minimizing the absorption of water. Hence, Kostiv et al. [73] designed the NaYF_4_:Yb^3+^/Er^3+^@NaYF_4_:Nd^3+^ core–shell NPs doped with Yb^3+^ and Nd^3+^ as sensitizers, and Er^3+^ as an activator for bioimaging. The core was uniform, with a thickness of 24 nm, whereas the core–shell particles had tuneable shell thicknesses of ∼0.5–4 nm. They were coated with in-house synthesized poly ethyleneglycol (PEG)-neridronate terminated with alkyne (Alk) to ensure their dispersibility in biological media. The stability of NaYF_4_:Yb^3+^/Er^3+^@NaYF_4_:Nd^3+^-PEG-Alk NPs in water or 0.01 M PBS, and the presence of PEG on the surface were determined. These Ln NPs were considered non-invasive probes for specific bioimaging of cells and tissues.

## 4. Nanoheterostructures Based on IHP NCs and Ln NPs 

In recent years, all-inorganic CsPbX_3_ (X = Cl, Br and I) perovskite NCs have proven to be promising materials in the field of optoelectronics due to their outstanding linear optical properties, even though the nonlinear properties of these perovskites are limited due to their small multiphoton absorption cross section and requirement of high-power density excitation. Interestingly, Zheng et al. [74] proposed a convenient strategy for tuning the UCL in CsPbX_3_ perovskite NCs through the sensitization of Ln^3+^-doped NPs. Particularly, CsPbX_3_ NCs and LiYbF_4_:0.5%Tm^3+^@LiYF_4_ core/shell Ln NPs were dispersed in cyclohexane to lead a homogeneous colloid. The CsPbX_3_-concentration-dependent UCL spectra for LiYbF_4_:0.5%Tm^3+^@LiYF_4_ core/shell NP-CsPbBr_3_ perovskite NCs were obtained under excitation at 980 nm, as depicted in Figure 5. Full-colour emissions with wavelengths beyond the availability of lanthanide ions were attained by adjusting the band gap of IHP NCs. These results presented the first panorama for photon UC with high efficiency, multiple colours and a tuneable lifetime of perovskite NCs under an excitation at low power density. It is important to note that the IHP NC luminescent lifetime was lengthened from the intrinsic nanosecond scale to milliseconds due to radiative energy transfer (RET) from Ln-doped NPs to IHP NC. This work opens a new avenue for the exploration of perovskite NCs based on UCL toward versatile applications such as ultrasensitive bioassay and high-resolution bioimaging.

Furthermore, there is a great need to develop heterostructured NCs based on inorganic perovskites. More clearly, although perovskite QDs have excellent optical properties, their biological applications have not been explored much because of their poor stability and the short penetration depth of UV light into tissues. The combination of perovskite QDs with Ln NPs has provided stable hybrid NCs, which are NIR excitable and emission tuneable. Hence, Ruan et al. [75] synthesized perovskite–Ln NP hybrid NCs composed of perovskite NCs with cubic phase and Ln NPs with hexagonal phase. The heterostructured CsPbBr_3_–NaYF_4_:Yb,Tm NCs were synthesized in one pot and consisted of cubic-phase CsPbBr_3_ QDs embedded in hexagonal-phase NaYF_4_:Yb,Tm NPs, which thus formed a watermelon structure with multiple seeds, and a cubic-phase NaYF_4_:Yb,Tm NP was used as an intermediate transition phase. The hybrid NCs emitted the characteristic green fluorescence of CsPbBr_3_ QDs under UV light and UV-blue fluorescence under NIR light excitation of NaYF_4_:Yb,Tm NPs, thus revealing the co-existence of both CsPbBr_3_ and NaYF_4_:Yb,Tm in the same structure. Moreover, a green fluorescence was obtained upon NIR excitation when the NaYF_4_:Yb,Tm phase absorbed NIR light and transferred the energy to the CsPbBr_3_ phase. This work opens a new way for synthesizing heterostructured NCs that could be applied to many other materials.

Recently, Shao et al. [76] reported the sensitized emission of CsPbI_3_ perovskite NCs after NIR excitation of CaF_2_:Yb^3+^/Ho^3+^ as hierarchical nanospheres (HNSs) in CsPbI_3_ and CaF_2_:Yb^3+^/Ho^3+^ nanocomposite structures. By introducing an appropriate proportion of Br ions into the perovskite, the luminescence was tuned between 695 nm and 655 nm, as depicted in Figure 6. Moreover, the lifetime of CsPbI_3_ emissions was lengthened to several milliseconds due to energy transfer from long-lived Ho^3+^ to CsPbI_3_ perovskite NCs. The stability of CsPbI_3_ NCs was enhanced in the composites, which kept 90% of its PL after 30 days. The composites were printed on flexible substrates for dual-mode fluorescent encryption anti-counterfeiting application and possessed excellent fluorescence under the excitation of both UV and NIR light. Moreover, the CsPbI_3_-CaF_2_:Yb^3+^/Ho^3+^ nanocomposites proved to be highly water-soluble, ultrastable and highly biocompatible in cell imaging applications. This work provides a new strategy for developing photon UC in perovskite NCs and a new trial for the development of multifunctional materials.

Although the halide perovskite nanomaterials with superior linear properties are greatly employed in optoelectronics and photonics, their strong multiphoton absorption only makes them prospective for bioimaging applications. However, the instability of perovskites in aqueous solutions limited their biological applications. Talianov et al. [77] demonstrated their fluorescence and UCL imaging in living cells using CsPbBr_3_ NCs with improved water resistance for at least 24 h after their coating as individual particles with various silica-based shells. The quality of phTEOS-TMOS@CsPbBr_3_ NCs was confirmed by HRTM and SEM, X-ray diffraction analysis, Fourier-transform infrared and energy-dispersive X-ray spectroscopies as well as fluorescence optical microscopy. phTEOS-TMOS@CsPbBr_3_ NCs have enhanced water stability, and consequently, they are of interest for several bioimaging applications.

In addition, it is important to note that, Estebanez et al., designed 1D-ordered nanostructures comprising Ln NPs and IHP NCs with open peapod-like shells, which were provided by a PbSO_4_ polymer for the first time [78]. The sensitized emission of IHP was achieved by NIR excitation of nearby Ln NPs. Ln NPs with a NaYF_4_ matrix doped with Yb and Tm or Er and with an inert shell of NaYF_4_, in the case of core–shell Ln NPs, and all-inorganic CsPbX_3_ NPs were selected for these studies. Interestingly, the lead sulphate shell enhanced the luminescence of core–shell Ln NPs in the polymers by ≈20 fold, which plays an important role in the efficiency of sensitized emission of LHNPs under NIR excitation of Ln NP-IHP NC co-polymers as well as in the chemical stability of IHP NCs in contact with water. In addition, the co-polymers were prepared as colloids and deposited as solid films on a glass substrate. The lifetime of sensitized IHP emission and emission efficiency entirely depended on irradiance and sample conditions. These co-polymers are promising candidates for manufacturing the photonic devices. Table 1 summarizes the various applications of Ln-doped and undoped IHPs by means of a UC process.

## 5. Other UC Luminescence Materials

Li et al. [88] designed superstructures comprising a metal-organic framework as the core and Nd^3+^-sensitized Ln NPs as satellites using an electrostatic self-assembly strategy. This double photosensitizer superstructure has a three-mode imaging function, including magnetic resonance, UCL and fluorescence, as well as an excellent anti-tumour effect under NIR excitation (at 808 nm) according to in vitro and in vivo experiments. Thus, the red blood cells did not deteriorate in the presence of the superstructure. Moreover, exposure of BALB/c mice to a 808 nm laser for 5 min demonstrated a lower temperature of the irradiated area, at about 42 °C, which did not result in damage to the mice. By contrast, the temperature of the irradiated area was raised to above 50 °C when using a laser excitation at 980 nm, and consequently, the mice skin was severely burned. Then, it can be proposed that the laser excitation at 808 nm is more adequate for biological applications since it produces a much weaker tissue thermal effect.

At the same time, Sun et al. [89] synthesized Ln^3+^-doped nanocomposites, specifically NaYF_4_:Yb^3+^,Er^3+^@NaYF_4_-Ce6@mSiO_2_-CuS nanohybrids for the applications of sensing and therapy, which provided temperature feedback in the phototherapy treatment (PTT) and were involved in photodynamic therapy treatment (PDT). NaYF_4_:Yb^3+^,Er^3+^@NaYF_4_ NPs were coated with mesoporous SiO_2_ combined with a Chlorin e6 (Ce6) photosensitizer, which can be excited by the red emission of Er^3+^ to lead to NaYF_4_:Yb^3+^,Er^3+^@NaYF_4_-Ce6@mSiO_2._ Then, the citrate-capped CuS (Cit-CuS) NPs as a photothermal conversion agent were attached to the composite surface. The temperature of the PTT site was monitored by recording the I_525_/I_545_ ratio of green emissions, as depicted in Figure 7. Based on the guidance obtained from spectral experiments, the dual-modal tumour therapy and real-time temperature monitoring were investigated both in vitro and in vivo, obtaining reasonable results. 

## 6. Conclusions and Perspective 

In this review article, the UC emission properties of Ln-doped inorganic perovskite NPs as UC materials have been discussed. We summarized the advances in the development of these materials among others UC nanomaterials relevant for biological applications. The major challenge for transforming UC nanotechnology into real-world applications is to enhance both the brightness and emission efficiency of Ln-doped NPs. 

It is important to note that the unique optical properties of Ln-doped NPs have attracted enormous scientific and technological interests. Intentional doping with higher concentration of Ln ions into different sections across a single Ln-doped NP has been explored to enhance the desirable optical properties as well as to introduce multifunctionality. Thus far, only spherical core@shell nanostructures have been reported to modulate the energy transfer, and hence, further investigations on heterogeneous one-dimensional structures, including rods, plates and dumbbells, are still needed. Importantly, controlled growth towards atomic precision is highly recommended for a clear understanding of these sophisticated energy transfer processes and the tuning of UC emissions. More clearly, the arrangement of dopants with higher concentrations into a host matrix along one direction could confine a direction of energy transfer and, consequently, may create new properties and enable novel applications.

The unique optical properties of highly doped LnIHPs have a great impact in the biological and biomedical fields. It is noteworthy that small-sized and bright Ln-doped NPs are needed to those applications, but owing to brightness issues, most of the currently developed Ln-doped NPs are relatively large (ranging between 20 and 50 nm). The synthesis of highly doped sub-10 nm level Ln-doped NPs with desirable emissions comparable with that of quantum dots and organic dyes is an extraordinarily challenging issue. Recently, the fine-tuning of Ln-doped NP with sizes below 10 nm has been achieved via homogeneous doping at high-doping concentrations. Nevertheless, the fabrication of sub-10 nm level Ln-doped NPs with heterogeneously doped core@shell nanostructures is challenging.

Despite the abovementioned advantages of Ln-doped NPs, their biological applications are limited due to their low water dispersibility. In this regard, modification of the NP surface is essential for improving their hydrophilicity and biocompatibility. There are mainly two types of modifiers: (i) organic surfactants, including cetyl-trimethyl ammonium bromide and ethylene diamine tetraacetic acid, which are mostly used as ligands to control both particle growth and stabilization against aggregation, and (ii) bifunctional polymers, such as polyvinylpyrrolidone, chitosanpolyethylenimine, polyacrylic acid sodium salt and polyethylene glycol, which are often used as chelating and stabilizing agents to render Ln-doped NPs hydrophilic and to provide functional groups for bio-conjugations. 

The surface molecules not only play a crucial role in the controlled synthesis of nanomaterials but also significantly alter the luminescence properties of nanomaterials. Therefore, the recent developments in dye-sensitized Ln-doped NPs and surface phonon-enhanced Ln-doped NPs in the thermal field have become a hot topic. Usually, Ln-doped NCs exhibit a narrow band and low absorption coefficient. Interestingly, organic dyes have more than 10 times broader absorption spectra and 10^3^–10^4^-fold higher absorption cross sections than that of Yb^3+^ sensitizer ions in Ln NPs. Therefore, despite photostability issues, the dye-sensitized UC system may enhance the UC performance. The phonons at the surface of a highly Yb^3+^-doped UC system can control thermal quenching and significantly enhance the UC brightness, particularly in sub-10 nm level NPs. Moreover, the surface plasmons on Ln-doped NPs produce greatly enhanced UC emissions, especially red emission due to the local field enhancement effect. However, further studies based on biological applications are still needed.

Furthermore, the hierarchical structures mostly possessed superior optical properties when compared with traditional NPs; specifically, core–shell composites of hydrophobic Ln-doped NPs encapsulated within a SiO_2_ layer have recently received extensive scientific and technological interest. The surface silica prevented NPs from flocculation and provides room for decoration with functional groups such as thiol, amino and carboxyl groups, which allowed for greater control in conjugation protocols even though precise control of both the thickness and uniformity of SiO_2_ layers is rather difficult and, hence, most Ln-doped NPs may be packed together in a single layer of SiO_2_, which will result in aggregation. In addition, the stability of organic capping on Ln-doped NPs is crucial for efficient luminescence in aqueous solutions. Interestingly, polysulfonate (PSS) capping on a Ln-doped NP surface has been shown to be useful for preventing NP disintegration in water and provided superior stability in a highly acidic medium. For comparison, the bare Ln NPs progressively disintegrated into their compositional ions and thus caused undesirable interference in chemical or biological environments. Additionally, the PSS capping layer can be further functionalized to lead new functional Ln-doped nanohybrids. The heterostructured NCs of perovskite with Ln NPs exhibited enhanced water stability, and consequently, they are of interest for several bioimaging applications. Interestingly, Ln-doped NPs with thin PSS coating and their functionalization with DNA have recently been reported. Both Ln-doped NPs and DNA preserved their full functionality, as demonstrated by Förster resonance energy transfer hybridization assays with Cy3-conjugated complementary DNA. Ratiometric FRET from Ln-doped NP to Cy3 demonstrated that these nanohybrids are able to quantify miR20a in the 0.01–10 × 10^−9^ M concentration range with a detection limit of 30 × 10^−12^ M (4.5 fmol of miR20a) [90]. 

While great progress has been made in the biomedical field based on Ln-doped NPs for the last few years, there remain issues that hinder the potential applications of Ln doped NPs as therapeutic and bioimaging agents. Specifically, nanotoxicology and safety assessments are the most essential studies for clinical applications. In vitro and in vivo toxicity assessments can be used to prove that Ln-doped NPs exhibit no obvious toxicity, but the effects of Ln-doped NPs on small animals used for longer duration as well as the interaction between Ln-doped NPs and the immune system are still unknown; moreover, the interaction between Ln-doped NPs and proteins in blood is still unclear. Therefore, much more systematic investigations are still needed. For the toxicity problem, lead-free perovskite could be a promising candidate as a future research direction.

## Figures and Tables

**Figure 1 nanomaterials-12-02130-f001:**
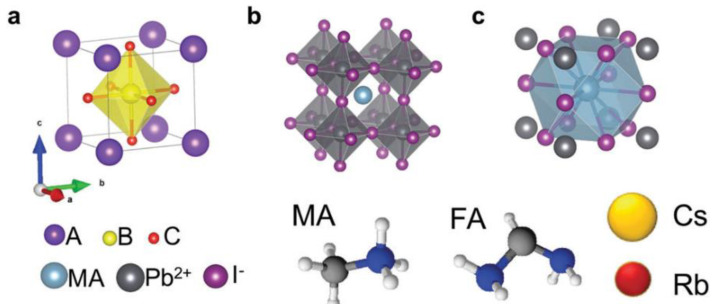
Structural features for metal halide perovskites. (**a**) Unit cell of general cubic perovskite; (**b**) MAPbI_3_ with octahedral coordination around lead ions; (**c**) MAPbI_3_ with cuboctahedra coordination around organic ions. Reprinted with permission [27]. Copyright 2017, Royal Society of Chemistry.

**Figure 2 nanomaterials-12-02130-f002:**
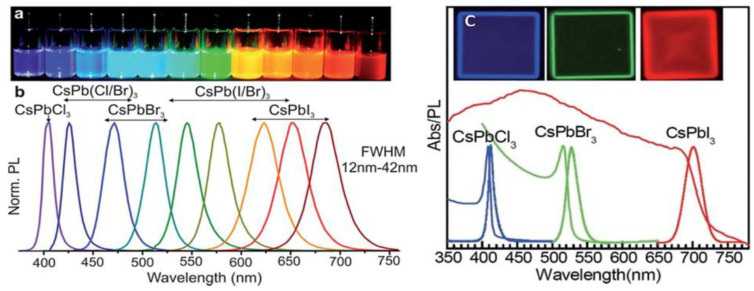
(**a**) Colloidal CsPbX_3_ NCs (X = Cl, Br and I) in toluene under UV lamp (λ = 365 nm); (**b**) corresponding PL spectra (λ_exc_ = 400 nm for all but 350 nm for CsPbCl_3_ NCs); (**c**) optical absorption, PL spectra and inset images for CsPbCl_3_, CsPbBr_3_ and CsPbI_3_ nanoplatelets. Reprinted with permission [49,50]. Copyright 2016, John Wiley & Sons and 2015, American Chemical society.

**Figure 3 nanomaterials-12-02130-f003:**
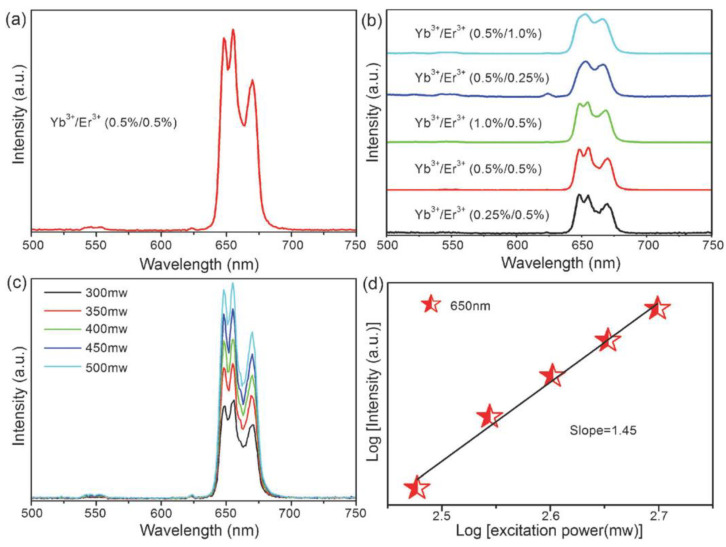
UC properties of KMgF_3_:Yb^3+^, Er^3+^ UCNCs. (**a**) A spectrum of single-band red UC emission; (**b**) UC emission spectra with various doping concentrations; (**c**) UC emission spectra with various excitation powers; (**d**) the corresponding logarithmic plot between UC intensity and excitation power. Reprinted with permission [62]. Copyright 2016, Royal Society of Chemistry.

**Figure 4 nanomaterials-12-02130-f004:**
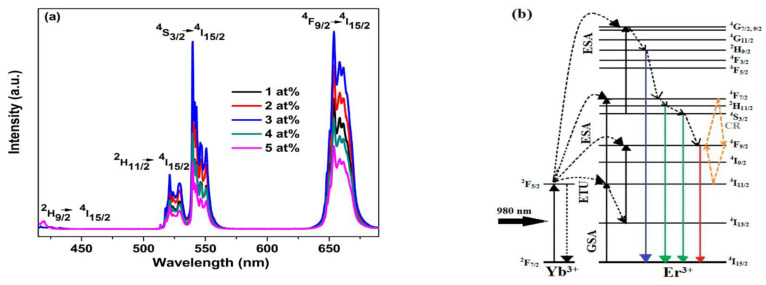
(**a**) UC emission of RbPbI_3_:Er^3+^ (10 at.%), Yb^3+^ with different doping concentrations of ytterbium under an excitation at 980 nm; (**b**) transition mechanism of UC process (RbPbI_3_:Er^3+^,Yb^3+^) at 980 nm excitation. Reprinted with permission [64]. Copyright 2018, American Institute of Physics.

**Figure 5 nanomaterials-12-02130-f005:**
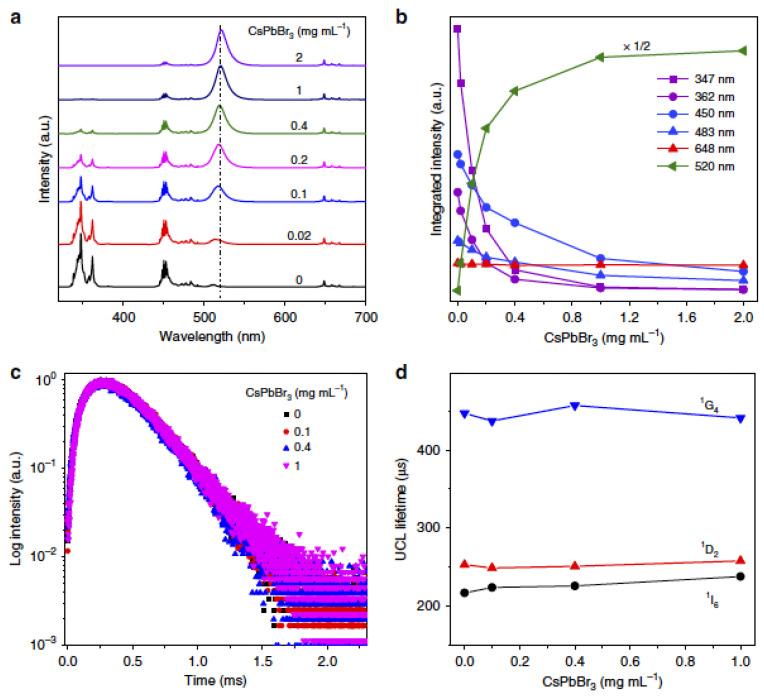
Investigation on the RET process in NP-sensitized CsPbX_3_ perovskite NCs. (**a**) CsPbBr_3_ concentration-dependent UCL spectra for LiYbF_4_:0.5%Tm^3+^@LiYF4 core/shell NP (1 mg·mL^−1^)-sensitized CsPbBr_3_ perovskite NCs excited at 980 nm; (**b**) integrated intensities for Tm^3+^ emissions and CsPbBr_3_ emission at 520 nm vs. the CsPbBr_3_ concentration from (**a**); (**c**) UCL decays from ^1^D_2_ of Tm^3+^ by monitoring the Tm^3+^ emission at 362 nm in NP-sensitized CsPbBr_3_ perovskite NCs with various concentrations excited at 980 nm; (**d**) UCL lifetimes of ^1^I_6_, ^1^D_2_ and ^1^G_4_ of Tm^3+^ in NP-sensitized CsPbBr_3_ perovskite NCs vs. the CsPbBr_3_ concentration. Reprinted with permission [74]. Copyright 2018, Nature Publishing.

**Figure 6 nanomaterials-12-02130-f006:**
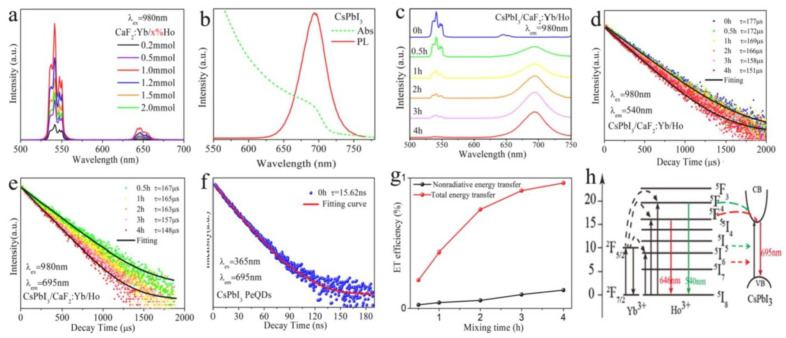
(**a**) UC emission spectra for CaF_2_:Yb^3+^ (20%)/Ho^3+^ (x%) HNSs; (**b**) absorption and excitonic emission spectra for CsPbI_3_ perovskite NCs; (**c**) UC emission spectra for HNS-perovskite NCs; dynamics of emissions for CsPbI_3_ perovskite NCs at (**d**) 540 nm, (**e**) 695 nm and (**f**) 695 nm; (**g**) calculated energy transfer efficiency of HNS-perovskite NCs with various times obtained from (**c**,**d**); (**h**) mechanism of UC emission in CsPbI_3_ and CaF_2_:Yb^3+^/Ho^3+^ composites. Reprinted with permission [76]. Copyright 2021, Elsevier.

**Figure 7 nanomaterials-12-02130-f007:**
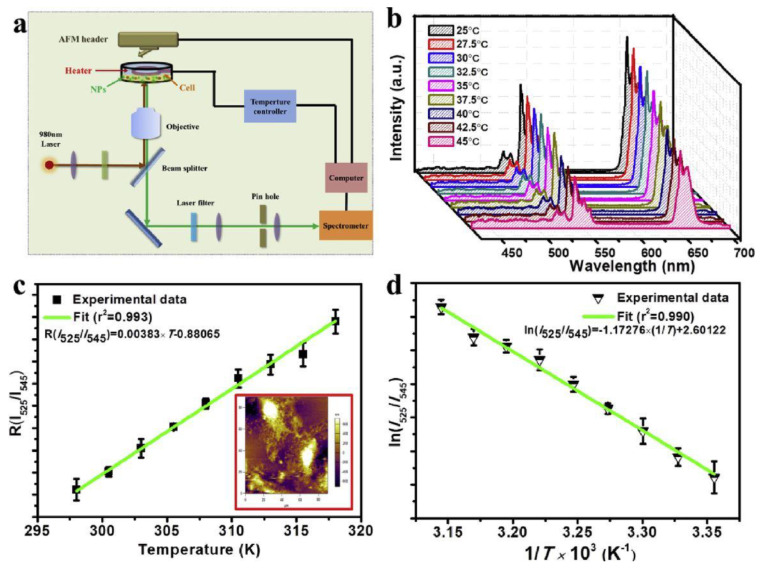
(**a**) Schematic diagram for the detection of temperature and emission spectrum of Ln NP-Ce6@mSiO_2_-CuS incubated with cells in physiological range; (**b**) UC emission spectra for Ln NP-Ce6@mSiO_2_-CuS incubated with cells at various temperatures by external heating; (**c**) finite impulse response (FIR) of green UC emissions for ^2^H_11/2_/^4^S_3/2_→^4^I_15/2_ transitions based on the temperature of Ln NP-Ce6@mSiO_2_-CuS incubated with cells (inset: AFM image of cell after spectral detection); (**d**) a plot of ln(I_525_/I_545_) versus 1/T to calibrate the thermometric scale for Ln NP-Ce6@mSiO_2_-CuS incubated with cells. Reprinted with permission [89]. Copyright 2019, Elsevier.

**Table 1 nanomaterials-12-02130-t001:** Lanthanide-doped inorganic halide perovskite NCs and non-doped perovskites for various applications.

S.No	Perovskites	Lanthanide Ions	Description	Application	Ref.
1	CsPbCl_3_ NCs	Ce^3+^, Sm^3+^, Eu^3+^, Tb^3+^, Dy^3+^ and Er^3+^	The introduction of lanthanide ions can considerably improve the PLQY of CsPbCl_3_ NCs and can provide visible light emissions and even NIR emissions.	Light emitting and other photoelectronic devices	[79]
2	CsPbCl_3_	Bi^3+^/Mn^2+^	The co-doped perovskite exhibits tuneable emissions spanning the wide range of correlated colour temperature (CCT) from 19,000 K to 4250 K under UV excitation. This interesting spectroscopic behaviour benefits from efficient energy transfer from the perovskite NCs to intrinsic energy levels of Bi^3+^ or Mn^2+^ doping ions.	Lighting and displays	[80]
3	CsPbCl_3_ NCs	Yb^3+^ and Yb^3+^/Er^3+^	The Yb^3+^-doped CsPbCl_3_ NCs emit strong NIR light at 986 nm, whereas the Yb^3+^/Er^3+^ co-doped CsPbCl_3_ NCs emit at 1533 nm. The total PLQY of the CsPbCl_3_ NCs changes from 5.0% to 127.8% upon incorporating 2.0% Yb^3+^, resulting in a 25.6 enhancement factor.	Diode lasers and photo-communications	[81]
4	CsPbX_3_ NCs	CaF_2_: Ln (Ln = Yb^3+^/Er^3+^, Yb^3+^/Ho^3+^ and Yb^3+^/Tm^3+^)	Owing to extremely high fluorescence resonance energy transfer (FRET) efficiency (~99.7%), excitonic UCL from CsPbX_3_ is performed under a low-power density of 980 nm diode laser irradiation.	Opto-electronics and photovoltaics	[82]
5	CsPbI_3_	NaYF_4_:Yb/Tm @NaYF_4_	An efficient single red band UC emission of CsPbI_3_ perovskite quantum dots (PQDs) was observed. In addition, the emission was easily regulated from 705 to 625 nm by introducing an appropriate proportion of Br ions, which is very difficult to achieve for traditional UCNPs. Moreover, benefiting from the efficient downshifting (DS) red emission of CsPbI_3_ PQDs, the composites displayed dual-wavelength excitation characteristics.	Dual-mode anticounterfeiting application	[83]
6	CsPbBrI_2_	w/o lanthanides	When photons only excite electrons in shallow trap states, some excited photons are absorbed by the shallow trap state, thus producing single-photon UCPL while the remaining photons are absorbed by the valence band, resulting in electron transfer from the valence band to the conduction band. Hence, the UC process is gradually dominated by a two-photon process as the energy of the incident photons decreases.	Optoelectronics	[84]
7	CsPbBr_1 × 2_ PQDs	NaYF_4_ Ln NPs	To improve the lattice matching between UCNPs and PQDs by replacing Y instead of Gd, the heterostructured CsPbBr_3_-NaGdF_4_:Yb,Tm NCs are obtained. They exhibit enhanced luminescence as well as stability at high temperatures, in polar solvents and under continuous UV excitation when compared with CsPbBr_3_-NaYF_4_:Yb,Tm nanocrystals and pure PQDs.	Optoelectronics	[85]
8	CsPbA_3_ (A = Cl, Br and I)	w/o lanthanides	An efficient UCPL with a striking phonon-assisted energy gain of ~8 kBT is obtained with high-quality, all-inorganic CsPbA_3_ perovskite NCs. In non-equilibrium conditions, the acoustic phonon UC recycles the population of optical modes and boosts the efficiency of photon UC.	Optoelectronics	[86]
9	CsPbBr_3_	w/o lanthanides	Vapor-phase epitaxial CsPbBr_3_ microplatelets are obtained with high crystallinity; self-formed high-quality microcavities; and great thermal stability, low-threshold and high-quality factor whispering-gallery mode lasing under one, two and three-photon excitation, and the lasing action is very stable under continuous pulsed laser irradiation (~3.6 Å~107 laser shots).	Lasing	[87]

## Data Availability

Not applicable.

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
