# Peer review of "Recent Progress in Lanthanide-Doped Inorganic Perovskite Nanocrystals and Nanoheterostructures: A Future Vision of Bioimaging"

_nanomaterials, 2022, doi:10.3390/nano12132130_

Round 1

Reviewer 1 Report

The paper is well structured, well written and surely deserves publication. I think it is a valid reviewed about the progress in lanthanide-doped perovskite. Surely interesting for a wide audience. English is ok but I am not a native speaker. I do not have any major concerns about the paper and suggest its publication in the journal.

Author Response

Thank you for your kind comments. 

Reviewer 2 Report

Dear Authors,

I have read your manuscript, entitled:  Recent progress in lanthanide-doped inorganic perovskite  nanocrystals and nanoheterostructures” and I have found it very interesting and well prepared.

The subject of your article is nowadays very topical because these type of materials have great potential not only in optoelectronic and photovoltaic but also  is biology for imaging and phototherapy. The structure of your work is very logical and well planned.

So I recommend your work for publication, however a minor revision should be done.

My main remarks:

·         As a Review Article this manuscript  seems to be too short with too small amount of references. There are many publications, within 5 last years, about Ln-IHP and their different properties and applications, for example:

 DOI: 10.1021/acs.nanolett.7b04575

https://doi.org/10.1088/2515-7655/abff18

 https://doi.org/10.1016/j.nanoen.2020.105354

·         I suggest to add the tables with collected results, obtain by different authors;

These tables will be useful for readers . (see the review article  Polymers 2022, 14, 1946. https://doi.org/10.3390/polym14101946)

·         In the index (first page) in the point 5, abbreviation UC should be explained, because is used for the first time.

Author Response

  1. As a Review Article this manuscript seems to be too short with too small amount of There are many publications, within last 5 years, about Ln-IHP and their different properties and applications, for example: DOI: 10.1021/acs.nanolett, https://doi.org/10.1088/2515-7655/abff18, https://doi.org/10.1016/j.nanoen.2020.105354.

Response: Thank you for your comment. We have added some of the publications you mentioned related to Ln-IHP in Table 1 together with their properties and applications in our revised version; they are highlighted in red.

  1.  I suggest to add the tables with collected results, obtain by different authors. These tables will be useful for readers. (see the review articles Polymers 2022, 14, 1946. https://doi.org/10.3390/polym14101946)

Response: Thank you for your suggestion. We have included the table related to Ln doped perovskites for various applications, again highlighted in red.

  1. In the index (first page) in the point 5, abbreviation UC should be explained, because is used for the first time.

Response: Thank you for your comment. The abbreviation has now been explained.

Reviewer 3 Report

In this manuscript, the authors have summarized the recent advances of Ln-doped all-inorganic halide perovskites (LnIHP), discussed the special optical properties of nano-composites based on the interaction between IHP NCs and Ln NPs and proposed the potential of future application. This manuscript is well organized to display a general impression of Ln NPs and LnIHP, which can be accepted for publication in Nanomaterials after a minor revision.

1          In introduction, the authors mentioned that “it is difficult to achieve nonlinear up conversion (UC) emission IHP NCs due to their low multiphoton efficiency and lack of intermediate energy levels”. However, it may be confusing for the readers. Thus, some discussions about the definition of UC emission and the difference between UC emission and normal emission of inorganic perovskites are required before this statement.

2          In page 2, line 55-57, more literatures about the biological applications of Ln NPs are required.

3          In page 2, line 62-64, the authors mentioned bio-applications of IHP NCs are limited by the lead toxicity. However, many examples about the combination of Ln NPs and Pb-based all inorganic perovskites are discussed. More discussions and explanations about the limitations of IHP NCs biological applications are required.

4          In page 3, line 87-91, the authors mentioned MA/FAPbX3 and all inorganic perovskites. However, before discussing the all-inorganic halide perovskites in the next section, some discussions about organic-inorganic halide perovskites are required. Some new literatures can be referred such as J. Mater. Sci. Technol. 113 (2022) 138-146; Adv. Mater. 33 (2021) 2002582; Chem. Eng. J. 420 (2021) 127599; Small 17 (2021) 2102186; J. Energy Chem. 62 (2021) 243-251.

5          The possibility of Ln doping in organic-inorganic halide perovskites such as MAPbI3 should be discussed.

6          Some new and relevant literatures about halide and oxide perovskites for solar-based applications should be cited and discussed such as Mater. Today Energy 23 (2022) 100896; Mater. Today Energy 23 (2022) 100899; Appl. Phys. Rev. 8 (2021) 041402; Adv. Funct. Mater. 30 (2020) 2001557; Angew. Chem. Int. Ed. 59 (2020) 136-152; ACS Appl. Mater. Interfaces 12 (2020) 23984-23994; Energy Fuels 34 (2020) 9208-9221; Energy Fuels 34 (2020) 10513-10528; etc.

Author Response

  1. In introduction, the authors mentioned that “it is difficult to achieve nonlinear up conversion (UC) emission IHP NCs due to their low multiphoton efficiency and lack of intermediate energy levels. However, it may be confusing for the readers. Thus, some discussions about the definition of UC emission and the difference between UC emission and normal emission of inorganic perovskites are required before this statement.

Response: Thank you for your comment. As per your suggestion, the definition of UC emission and the difference between normal and upconversion emission are now included in the revised version and it is highlighted in red:

“IHP NCs usually exhibit linear optical properties under UV and visible light. It is important to note that near-infrared (NIR) photon upconversion is a unique process which converts two or more NIR photons into one of a higher energy, thus leading to anti-Stokes emission. It is very difficult to achieve nonlinear upconversion (UC) emission in IHP ma-terials because of their low multiphoton efficiency and lack of intermediate energy levels.”

  1. In page 2, line 55-57, more literatures about the biological applications of Ln NPs are required.

Response: Thank you for your suggestion. We have added more references in line 51-52, highlighted in red.

  1. In page 2, line 62-64, the authors mentioned bio-applications of IHP NCs are limited by the lead toxicity. However, many examples about the combination of Ln NPs and Pb-based all inorganic perovskites are discussed. More discussions and explanations about the limitations of IHP NCs biological applications are required.

Response: Thank you for your suggestion. More discussion about limitations of Pb-based all inorganic perovskites with Ln NPs in biological applications has been included in the revised manuscript which is highlighted in red and given below:

“However, the superior optical properties of IHP NCs provide several advantages to be used in the field of bioimaging. Nevertheless, there are many issues, such as the stability of these materials under physiological conditions, their integration in biomolecules during circulation, epigenetic interaction, and possible toxicity during degradation pro-cesses, to be clarified before real life implementation.”

  1. In page 3, line 87-91, the authors mentioned MA/FAPbX3 and all inorganic perovskites. However, before discussing the all-inorganic halide perovskites in the next section, some discussions about organic-inorganic halide perovskites are required. Some new literatures can be referred such as J. Mater.Sci. Technol. 113 (2022) 138-146; Adv. Mater. 33 (2021)2002582; Chem. Eng. J. 420 (2021) 127599; Small 17 (2021)2102186; J. Energy Chem. 62 (2021) 243-251.

Response: Thank you for your comment. In accordance, we have included some sentences about organic-inorganic halide perovskites in the revised manuscript which are highlighted in red and given below.

“This is based on their superior properties: tuneable bandgap, high light absorption capability, long carrier diffusion length and inexpensive raw materials for their preparation. However, they have been affected by instability issues not only in solar cells but also in other applications, such as in LEDs and bioimaging. Inorganic perovskites have been introduced in optoelectronic devices to overcome these issues.28-30 Inorganic perovskite oxide, lead-free inorganic perovskites and inorganic perovskites with mixed halides have also been developed in the past decades.31-38”.

  1. The possibility of Ln doping in organic-inorganic halide perovskites such as MAPbI3 should be discussed.

Response: Thank you for your valuable suggestion. We have added some sentences regarding the possibilities of Ln doping in organic-inorganic halide perovskites in section 2.2 which is highlighted in red and given below:

“Ln doping in hybrid perovskites also plays a key role in their optoelectronic properties. For example, MAPbI3 with lanthanides has been considered as a desirable candidate for improving the permanence of perovskite solar cells. The incorporation of various lanthanide ions (Ln3+ = Ce3+, Eu3+, Nd3+, Sm3+, or Yb3+) into perovskite films largely enhance their performance, such as the efficiency and stability of the device, which is attributed to an enlarged grain size and crystallinity [20].”

  1. Some new and relevant literatures about halide and oxide perovskites for solar-based applications should be cited discussed such as Mater. Today Energy 23 (2022) 100896; Mater. Today Energy 23 (2022) 100899; Appl. Phys. Rev. 8(2021) 041402; Adv. Funct. Mater. 30 (2020) 2001557; Angew. Chem. Int. Ed. 59 (2020) 136-152; ACS Appl. Mater. Interfaces12 (2020) 23984-23994; Energy Fuels 34 (2020) 9208-9221; Energy Fuels 34 (2020) 10513-10528; etc.

Response: Thank you for your suggestion. The above-mentioned publications have been cited in the Introduction of the revised manuscript and are highlighted in red.

Reviewer 4 Report

To date, more than 200 reviews have been published on methods for synthesizing and controlling the optical properties of perovskites. For this reason, the authors should more clearly formulate in the title the area of ​​their possible application discussed in the review (bio-imaging?). A significant number of works on the topics discussed in the review were published in 2021-2022. This should be taken into account in the list of publications.

 In general, the review is quite original and may be of interest to specialists. The article corresponds to the subject of the journal and can be published after taking into account the above comments.

Author Response

To date, more than 200 reviews have been published on methods for synthesizing and controlling the optical properties of perovskites. For this reason, the authors should more clearly formulate in the title the area of their possible application discussed in the review (bio-imaging?). A significant number of works on the topics discussed in the review were published in 2021-2022. This should be taken into account in the list of publications.

In general, the review is quite original and may be of interest to specialists. The article corresponds to the subject of the journal and can be published after taking into account the above comments.

Response: Thank you for your  valuable suggestions. The title has been modified in the revised manuscript. It now reads “Recent progress in lanthanide-doped inorganic perovskite nanocrystals and nanoheterostructures: a future vision of bioimaging

Moreover, we have included recent publications (2021-2022) in Table 1 of the revised manuscript and are highlighted in red.